# The Immune Microenvironment in Penile Cancer and Rationale for Immunotherapy

**DOI:** 10.3390/jcm9103334

**Published:** 2020-10-17

**Authors:** Mohamed E. Ahmed, Shayan Falasiri, Ali Hajiran, Jad Chahoud, Philippe E. Spiess

**Affiliations:** 1Department of Urology, Mayo Clinic, Rochester, MN 55905, USA; 2Department of Urology, University of South Florida Morsani College of Medicine, Tampa, FL 33612, USA; falasiri@usf.edu; 3Department of Genitourinary Oncology, H Lee Moffitt Cancer Center and Research Institute, Tampa, FL 33612, USA; Ali.Hajiran@moffitt.org (A.H.); Jad.Chahoud@moffitt.org (J.C.); Philippe.spiess@moffitt.org (P.E.S.)

**Keywords:** penile cancer, tumor microenvironment (TME), immune infiltration patterns, Programmed Death-1 Ligand (PD-L1), tumor mutational burden (TMB)

## Abstract

Penile cancer is an extremely rare malignancy that accounts for approximately 1% of cancer deaths in the United States every year. While primary penile cancer can be managed surgically, advanced and metastatic forms of the disease require more aggressive management plans with systemic chemotherapy and/or radiotherapy. Despite the meaningful response to systemic treatments, the 2-year progression-free survival and disease-specific survival have shown disappointing results. Therefore, there is a crucial need for alternative treatment options with more favorable outcomes and a lower toxicity profile. There are currently extensive studies of tumor molecular biology and clinical trials with targeted molecular therapies, such as PD-1, PD-L1, and CTLA-4. In this review, we will describe the penile cancer microenvironment, and summarize the rationale for immunotherapy in penile cancer patients.

## 1. Introduction

Penile cancer (PeCa) is an exceedingly rare malignancy. In 2020, it is estimated that penile cancer will comprise 0.25% of male cancer incidence and account for 440 deaths in the United States [1]. Geographic disparities are pronounced between Western countries and the developing world, where incidence may run as high as 2.8–3.7 per 100 000 [2]. Globally, penile cancer accounts for 34,475 new cancer cases and 15,138 cancer deaths every year [3]. The vast majority (~95%) of penile cancers is represented by squamous cell carcinoma (SCC) [4,5]. Known risk factors for penile SCC include phimosis, smoking, chronic inflammatory states, a high number of sexual partners, lack of circumcision, and human papillomavirus (HPV) infection [6,7,8,9]. While penile SCC has a separate entity of SCCs and treated differently, there are recent reports describing extensive similarities and commonalities in the genetic and pathogenesis regulators of SCCs of various sites, including both the general determinants in the cancer process, such as P53 and cyclin D1, or the specific regulators, such as NOTCH, SOX2, and TP63 genes [10]. 

Though penile SCC is an active area of research, therapeutic options are limited, with 5-year overall survival rates of 63% and 70% in the United States and Europe, respectively [11,12]. The extent of lymph node metastases at inguinal node dissection most strongly predicts prognosis as there are few effective therapies when the regional disease is present. Stratified by lymph node status, 5-year overall survival is >85% for patients with negative nodes and 29%–40% for patients with positive nodes [13]. Even for patients with clinically negative groins (cN0), the likelihood of metastasis approaches 25% [14]. Surgery remains the cornerstone of penile cancer treatment for both primary tumors and lymph node metastasis (LNM) with multimodal treatment, in the form of neoadjuvant chemotherapy or adjuvant radiation, reserved for advanced stages [15]. In regards to systemic therapy, while 50% of the patients achieve an objective response to taxane-ifosphamide-platinum regimens, the majority have demonstrated disappointing 2-year progression-free survival (PFS) and disease-specific survival (DSS) probability of 12% and 28%, respectively [16,17,18]. Furthermore, Wang et al. reported poor survival outcomes of salvage treatment for disease recurrence after first-line chemotherapy, with a median overall survival of fewer than six months [19]. Thus, there is much interest in developing novel strategies with higher efficacy and low toxicity profile.

The discovery of Immune checkpoint inhibitors (PD1 / PD-L1 and CTLA-4) and the use of monoclonal antibody represent a revolutionary step in the management of many cancers. In an open-label phase 3 trial, treatment with pembrolizumab versus methotrexate, docetaxel, or cetuximab for recurrent or metastatic head-and-neck squamous cell carcinoma showed a favorable safety profile and prolonged overall survival [20]. The current advances in immunotherapy, along with the reported spectacular therapeutic outcomes, have inspired physicians to investigate the feasibility of its use in many cancers, including penile cancer. In this article, we review the microenvironment of penile carcinoma and provide a justification for immunotherapy use in these cases.

## 2. Immune Infiltration Patterns (CD8, FOXP3 T regs)

Tumor-infiltrating lymphocytes (TILs) have been frequently studied for their roles in triggering the host immune response to many forms of cancer, as well as in the processes of cancer immunoediting and immune escape [21]. Immunologically, tumors can be divided into three subgroups according to the number of intraepithelial and stromal cytotoxic T lymphocyte: (1) “Immune desert”; (2) “immune excluded”; and (3) inflamed [22,23,24]. The populations of lymphocytes compromising these infiltration patterns further stratify with ongoing investigations into CD3^+^/CD4^+^ T cells, CD3^+^/CD8^+^ T cells, and CD3^+^/CD4^+^/FOXP3^+^ T_reg_ cells (Figure 1) [25,26]. In pancreatic cancer, the ratio of T_regs_ to CD4^+^ T (%T_reg_) has demonstrated a significant association with shorter survival, while tumor-infiltrating CD4^+^ T^high^/CD8^+^ T^high^/%T_reg_^low^ independently predicted longer overall survival [26]. The IMvigor210 study provided evidence associating CD8^+^ density within the tumor with favorable atezolizumab response in metastatic urothelial cancer [27]. Thus, higher levels of CD8^+^ T cells is correlated with better prognosis in many cancers, including squamous cell carcinomas and urologic malignancies, however, there is growing evidence that sub-set populations of these CD8^+^ T cells are exhausted and lack their cytolytic activity, as well as the production of effector cytokines leading to impaired the antitumor activity [28,29].

Specifically, in penile SCC, the immune infiltration patterns have generated considerable interest in further dissecting out this complexity. A recent study characterized the immune microenvironment in 54 patients with penile SCC using IHC with the immune markers: CD3, CD8, CD68, PD-1, PD-L1, Pancytokeratin, and DAPI. Notably, this cohort was analyzed for the effect of the exhausted, cytotoxic T cell population sub-type (CD3^+^/CD8^+^/PD-1^+^), demonstrating that high densities of stromal cytotoxic, antigen-experienced T cells, suggestive of an immune excluded type, were significantly associated with worse median OS (27 vs. 102 months *p* = 0.05) [30]. Another study by Ottenhof et al. in 2018 offered evidence that low stromal CD8^+^ T cell was associated with LNM [23]. In a 2015 study by Vassallo et al., penile SCC with high levels of tumor-infiltrating FOXP3^+^ T_reg_ cells bore a worse disease-free survival probability (HR 2.50, *p* = 0.02) [31]. 

## 3. PD-L1 Expression

Programmed death-ligand 1 (PD-L1) has been detected in 40-60% of penile SCC and mainly high-risk HPV negative tumors [32]. In many cancers, increased PD-L1 expression by either the tumor cells or the host immune cells, especially tumor-associated macrophages (TAMs), has correlated with poor prognosis and lower numbers of TILs [33]. Nearly two-thirds of primary penile SCC tumors are PD-L1-positive, with PD-L1 positivity defined by >5% tumor expression [32]. In an immune histochemical study of 37 penile SCC, Udageret al. reported that PD-L1 expression in the primary tumor showed a significant association with regional lymph node metastasis (LNM, *p* = 0.024), as well as shorter cancer-specific survival (CSS, *p* = 0.011) [32] Additionally, In a multivariable analysis of 213 penile SCC patients, Ottenhof et al. reported that only diffuse PD-L1 expression in tumor cells was a significant predictor of lymph node metastases with OR of 2.81 (*p*-value = 0.05). Furthermore, high-risk human papilloma virus-negative status (hrHPV) and diffuse PD-L1 expression in the tumor field demonstrated significant correlation with poor disease-specific survival with an HR of 9.73 (*p*-value < 0.01), and HR of 2.81 (*p*-value=0.03), respectively [23]. Moreover, in a recent case study, Trafalis et al. reported a partial response to nivolumab, an anti-PD-1 monoclonal antibody, in a patient with advanced hrHPV negative penile SCC refractory to chemoradiation therapies [34]. The authors reported a >80% reduction in tumor volume after eight cycles of nivolumab. Of note, the patient pre-treatment histology showed >5% expression of PD-L1, while post-treatment histology on residual tumor cells revealed attenuation of PD-L1 expression with significant augmentation of PD-L1 expression on immune cellular elements surrounding tumor cells, suggesting the use of combination therapy with an anti-PD-1/PD-L1 agent. 

Ongoing penile SCC-related phase 2 clinical trials, targeting PD-L1/PD-1, include NCT02834013, NCT03333616, NCT03074513, and NCT02721732 [35,36,37,38]. NCT02834013 employs both nivolumab and ipilumumab in patients with rare tumors. NCT03333616 studying the combination of nivolumab and ipilumumab for advanced rare genitourinary tumors. NCT03333616 hopes to study how well atezolizumab and bevacizumab work in treating patients with rare solid tumors. NCT02721732 is evaluating the efficacy of pembolizumab in patients with rare tumors [38]. Table 1 summarized the most recent and ongoing clinical trials in penile cancer patients, as per 2020 updates.

## 4. Macrophage/MDSC Infiltration Patterns 

Tumor-associated macrophages (TAMs) perform a major role in the tumor microenvironment by increasing angiogenesis, enhancing tumor cell mobility, and modulating immunotolerance. Both TAMs and TGF-B are found to be associated with VEGF, the expression of which has been demonstrated to be an independent prognostic factor for metastatic progression in penile carcinoma [39]. Moreover, TAMs have been implicated in establishing pre-metastatic niches, which may have a role as a tumor survival mechanism against systemic chemotherapy [35,40]. 

The continuously evolving paradigm divides TAMs into M1 (classical) macrophages CD68^+^/CD86^+^/HLA-DR^+^ and M2 (alternative) macrophages CD163^+^/CD206^+^/HLA-DR^+^ [41,42]. Studies in cancer patients have associated cancer progression and worse overall survival with the M2 phenotype [26]. In cervical SCC, M2-polarized TAMs have been identified to express PD-L1 [43]. In the context that PD-1 is found on most infiltrating CD8 + T cells and the evidence supporting PD-1/PD-L1 interactions as a driving factor behind cancer immune tolerance, these findings suggest that M2 TAMs may contribute to cancer immune escape [44]. In cervical SCC, M2 macrophage differentiation has been significantly linked to cancer-derived IL-6 and prostaglandin E2 [41]. Remarkably, M2 macrophages might be stimulated to classical tumor-rejecting M1 macrophages via CD40 interactions in the presence of IFN-gamma [41]. Additionally, IL6/JAK/STAT3 signaling pathway has been proposed as one of the mechanisms that can affect tumor microenvironment and immune response to tumor cells [45]. IL-6 activation induces JAK/STAT3 pathway in both tumor cells and tumor-infiltrating immune cells. Subsequently, myelomonocytic infiltrating tumor cells differentiate into impaired dendritic immune cells producing protumorigenic matrix-metalloproteinase (MMP-9) or M2 macrophages expressing PD-L1 that, in turn, suppresses cytotoxic T cell response [45,46]. Therefore, recent data suggests that IL-6/JAK/STAT3 signaling pathway may represent a therapeutic target to suppress tumor growth and activate the antitumor immune response [47,48]. However, in phase 2 clinical trials, the use of an anti-IL-6 monoclonal antibody in metastatic castration-resistant prostate cancer and in advanced solid tumors didn’t show clinical benefits [49,50]. 

In penile SCC, high densities of CD68^+^ TAMs were associated with significantly improved cancer-specific survival (CSS) (*p* = 0.04), overall survival (OS) (*p* = 0.02), and lower risk of regional recurrence (*p* = 0.04) [30]. Another penile SCC group detected high intra-tumoral CD163^+^ corresponding with LNM [23]. 

In addition to that, Myeloid-derived suppressor cells (MDSC), believed to be immature bone marrow-derived hematopoietic progenitor cells failing, identified by CD11b^+^/CD33^+^/HLA-DR^−^, and expressing several functional markers, such as arginase, represent a complex constituency of the TME with the ability to suppress T cell responses [51,52,53]. Monocytic MDSC (M-MDSC) [CD11b^+^/CD33^+^/HLA-DR^−^/Lys6C^+^/CD14^+^], and polymorphonuclear neutrophils MDSC (PMN-MDSC) [CD11b^+^/CD33^+^/HLA-DR^−^/Lys6G^+^/CD15^+^] represent the major constitutes of MDSCs [54,55]. MDSCs are induced in both inflammatory and cancerous conditions. Factors induced MDSC to include PGE2, IL-1β, IL-6, VEGF, and C5a.complement component [56]. This could explain the relationship between chronic inflammation and progression to cancer. In a genetically engineered mouse model of penile SCC, huang et al. described changes in tumor microenvironment with a marked reduction in CD-8^+^ T cells, NK cells, B cells, and tumor-associated macrophages, while there was a marked increase in CD-11b ^+^ with its ability to suppress the proliferation of the cytotoxic T cells [57]. 

Several studies have reported that tumor-infiltrating-MDSCs suppress the immune system by producing arginase, resulting in depletion of L-arginine in the tumor microenvironment and suppression of T-cell response [58,59]. Of note, the arginase enzyme has many isoforms; the cytosolic arginase-1 and the mitochondrial arginase-2 enzyme. Arginase expression by MDSC is induced by several factors, including immunosuppressive cytokines (TGF-β, IL-4, IL-10, and IL-13), tissue hypoxia, and acidosis [60]. Recently, there is increasing evidence linking the expression of arginase-1/2 with poor prognosis in several cancers, including head and neck cancer, pancreatic cancer, and acute myeloid leukemia [61,62,63]. However, expression of arginase in genitourinary cancers has not yet been described—particularly prostate cancer and renal cell carcinoma [64,65,66]. 

The presence of MDSCs in the TME contributes to tumor-mediated immune escape and bear a negative correlation to OS [53]. 

The expansion of MDSCs may be suppressed in many instances. In HNSCC, tadalafil, in its function as a phosphodiesterase-5 (PDE5) inhibitor, has demonstrated significant immunomodulation of the TME by lowering MDSCs and T_regs_, while increasing tumor-infiltrating CD8^+^ T cells in a dose-dependent fashion [55,67]. In RCC and prostate cancer, sunitinib, as a VEGF, PDGFR, and receptor tyrosine kinase inhibitor, has demonstrated promise in the reduction of MDSCs [68,69]. Additionally, in a genetically engineered mouse (GEM) model of pSCC, the use of cabozantinib or celecoxib has exhibited a synergistic effect with immune checkpoints inhibitors (ICB) by lowering MDSCs positive for CD-11b^+^ and Ly6G^+^ [57]. 

Moreover, cabozantinib is a small molecular inhibitor of the Tyrosine kinases c-Met and VEGFR2 that has been shown to increase MDSC CD40^+^ expression compared to baseline in metastatic urothelial carcinoma (*p* = 0.0005) [70]. CD40 activation of dendritic cells in the TME has, in turn, been linked to increased T cell-mediated immunity [71].

Currently, there are ongoing trials to test the efficacy of combination immunotherapies in advanced genitourinary cancers, including penile SCC. Examples include aphase ½ clinical trial (NCT03866382) to investigate the efficacy of the combination of nivolumab and ipilimumab in addition to cabozantinib in rare genitourinary tumors [56]. Additionally, there is a phase 1 clinical trial (NCT02496208) investigating cabozantinib and nivolumab with or without ipilimumab in metastatic genitourinary tumors [69]. 

## 5. HPV Role in Tumor Immune Microenvironment

The relationship of HPV to the penile SCC TME represents an area of active interest for immunotherapy augmentation. Though there are variations in the prevalence of HPV that are primarily attributed to differences in sampling, viral molecular testing, and population studied, a systematic review of 1266 invasive penile SCC patients reported that in North America, up to 48.7% of penile SCC harbor HPV DNA. The vast majority of HPV-positive cases were represented by the high-risk HPV (hrHPV) 16 and 18 subtypes, 30.8% and 6.6%, respectively [5]. Stratified by HPV status, hrHPV negativity in 213 penile SCC correlated significantly with poor disease-specific survival (HR 9.7, *p* < 0.01) [23]. Viral positivity in penile SCC has repeatedly demonstrated favorable outcomes in survival, which may be linked to a theoretical increased production of neo-antigens [72]. However, the detection of HPV infection is not enough evidence of HPV-induced cancer. Therefore, there has been a growing interest in discovering new markers that can prove transcriptionally active infections [73]. Zargar-Shoshtari et al. investigated the potential clinical utility of two common tumor proteins, p16 and p53, in the HPV pathway [74]. The authors reported that patients with negative p16 and positive p53 are at increased risk of nodal metastases (OR: 4.4, 95% CI: 1.04–18.6). Also, they reported that positive p16 status was associated with longer cancer-specific survival (HR: 0.36, 95% CI: 0.13–0.99), with the worst CSS was seen in patients with lymph node-positive disease, as well as double negative of p16 and p53 [74]. Moreover, Aziz et al. investigated the prognostic value of the upregulation of PI3K-AKT-mTOR signaling pathway in penile SCC [75]. The authors reported that increased expression of PI3K-AKT-mTOR was associated with a lower risk of recurrence and overall mortality. They suggested that the use of mTOR pathway biomarkers with HPV infection status may demonstrate a prognostic value that can help in the risk stratification of patients with penile SCC [75]. 

In an IHC study, HPV^+^ cases contained significantly higher stromal cytotoxic (CD8^+^) T cells than HPV^−^ cases (*p* = 0.04), representing a significant effect of HPV on the penile SCC TME [30]. Moreover, Lyford-Pike et al. reported that 70% of HPV-associated head and neck SCC demonstrated positive expression of the immunosuppressive molecule PD-L1 on both tumor cells and CD-68+ TAMs [76]. It has been reported that the majority of CD-8+ Tumor-infiltrating lymphocyte in HPV- associated head and neck SCC express PD-1, suggesting the benefits of using immune checkpoint blockades in these patients [76,77].

In light of the increasing relevance of HPV to penile SCC management, the European Association of Urology penile cancer 2018 update now recommends that the pathological evaluation of penile carcinoma specimens must include: (1) An assessment of the HPV status; (2) a diagnosis of the SCC subtype; and (3) an assessment of surgical margins including the width of the surgical margin [15].

There are several promising clinical trials investigating combinatorial immunotherapy augmented with HPV-targeted vaccines in HPV-associated malignancies. The phase ½ trial NCT04432597 is testing an HPV vaccine in combination with an anti-PDL1/TGF-Beta Trap drug. The phase 2 trial NCT03439085 is testing an HPV vaccine with durvulumab, a human immunoglobulin G1 kappa monoclonal antibody targeting PD-L1. The phase 2 trial NCT03427411 is studying the efficacy of M7824, a bifunctional fusion protein targeting TGF-β and PD-L1, for which phase 1 data has already demonstrated an encouraging safety profile and efficacy potential [78]. Another phase ½ trial NCT04287868 is utilizing a combination of M7824 and vaccinations in patients with advanced HPV-associated malignancies [79]. 

## 6. Tumor Mutational Burden

Tumor Mutational Burden (TMB) was described as the number of nonsynonymous somatic mutations in the coding area of tumor cells per megabase (MB) of DNA. These somatic mutations influence the expression of tumor-specific epitopes (neoantigens) that are targetable by the host immune system [80]. TMB has been described in tumors with mismatch repair defect (MMR), or microsatellite instability (MSI) defect, such as colorectal cancers, or defect in DNA replication [81,82]. However, the prevalence of TMB varies widely across tumors, with non-small cell lung cancer (NSCLC) demonstrates the largest number of these mutations (0.1–100 mut/Mb) [83,84]. The reported wide variance in the prevalence of TMB is also partly due to the lack of standardization of TMB quantification and reporting system. 

TMB has been suggested as a potential biomarker of patients’ prognosis. It has been hypothesized that increased expression of TMB indicates increased expression of PD-L1 and durable response to immune checkpoints inhibitors [85]. Recently, the Food and Drug Administration (FDA) has approved the use of pembrolizumab in patients with TMB ≥10 mutation/megabase [86]. This was based on the data from the non-randomized, open-label KEYNOTE-158 trial. They reported an objective response rate of 29% (95CI: 21%–39%), with approximately 50% of the responses were greater than two years [86,87]. 

Cancers with mismatch repair (MMR) deficiency, which invariably contributes to very high mutation rates, have consistently responded to pembrolizumab (PD-L1 blockade) [88]., Additionally, atezolizumab, an engineered humanized immunoglobulin G1 monoclonal antibody for PD-L1, has shown tolerability and durable activity in urothelial cancers with high TMB in proportion to increased levels of PD-L1 expression on immune cells [89]. Moreover, Van Allen et al. studied the relationship between response to CTLA-4 blockade and the genetic sequencings and mutations [90]. They reported that TMB and expression of cytotoxic cells in the tumor microenvironment were significant predictors of response to CTLA-4. 

In addition to that, IFN-gamma related genes were also associated with durable response to immunotherapy. In a study of top genes associated with response to pembrolizumb in patients with metastatic melanoma, the preliminary IFN-gamma 10 gene signatures were significantly associated with good response and were able to predictor responders from non-responders to pembrolizumab [91]. These genes included IFNG, STAT1, CCR5, CXCL9, CXCL10, CXCL11, IDO1, PRF1, GZMA, and MHCII HLA-DRA [91].

However, some chromosomal mutations are associated with poor response to immunotherapy. Davoli et al. reported that tumor aneuploidy or somatic copy number alterations (SCNAs) was associated with poor response to immunotherapy [92]. In addition, Xiao et al. studied in metastatic melanoma patients the relationship between TP53 mutation and the response to anti-CTLA-4 therapy. The authors reported that increased TP53 expression is significantly associated with poor response, poor progression-free survival, and poor overall survival, suggesting it as a negative predictor for response to CTLA-4 blockade [93].

Specific somatic mutations, copy number alterations, and gene and miRNA expression patterns have been demonstrated to be significantly associated with shorter time to progression or decreased survival. A 2018 study of 25 penile SCC patients treated with 1^st^-line cisplatin-based chemotherapy studied the expression of 738 genes. In univariate analysis, upregulated MAML2 (*p* = 0.004), KITLG (*p* ≤ 0.0001), and JAK1 (*p* = 0.029) genes were associated with poor overall survival. In contrast, genes, such as upregulated FANCA, were associated with better overall survival (*p* = 0.024) [94].

Despite the significant correlation of specific genetic alterations and expression patterns to penile SCC prognosis, a study of 27 whole-exome sequenced penile SCC tumors, investigating 810 genes, could not demonstrate an association between overall mutational burden and tumor stage, grade, or age though this relationship remains to be further explored in relation to immunotherapy response [95]. 

Feber et al. studied the genetic alteration in penile squamous cell carcinoma patients. Interestingly, the authors reported that tumors with high HPV viral load have a lower tumor mutational load than HPV negative tumors. This, in combination with the identification of a strong CpG signature in HPV-negative tumors (*p* < 0.001), suggests that studies of changes to the epigenome rather in addition to direct genetic alteration, may unveil key targets in the development of penile SCC [95]. Of note, Genetic studies of penile SCC have demonstrated at least 30% of cases exhibited a targetable gene alteration with many similarities with other SCCs, suggesting similar biology and potential targeting agent [96]. 

Additionally, recent reports have described the upregulation of Sox2 and Ptgs2 transcriptional factors in skin and penile SCC [57,97]. Sox2 is a β-catenin transcriptional target, upregulated in cancer stem cells, and absent in normal epidermis, while Ptgs 2, or better known as Cox2, is a pro-inflammatory gene that induces prostaglandins production from arachidonic acid resulting in cytokine upregulation and inflammatory reactions. These genes act as the master regulators of cancer stem cells that play an important role of Sox-2 in tumor initiation and progression. Boumahdi et al. reported that conditional deletion of Sox2-gene results in tumor cell regression, consistent with their critical role for disease continuity [97].

## 7. Conclusions

Penile squamous cell carcinoma is a complex disease with a debilitating nature. However, the current revolution and advances in immunotherapies hold promises for cancer patients, including penile cancer. Despite the lack of level1 evidence for using immunotherapy in penile cancer, most penile squamous cell carcinoma expresses PD-L1, which provides a rationale for considering immunotherapy. There are ongoing trials studying the use of a combination of anti-PD-1 and anti-CTLA-4 in rare Genitourinary cancers. Additionally, Preclinical data suggested synergistic efficacy with using combination therapy of Immune Checkpoints blockades (ICB) with MDSC-inhibiting agents, such as cabozantinib or celecoxib, particularly in the setting of chemoresistant disease. Moreover, the current advances in the understanding of tumor microenvironment and tumor mutational burden could help to select patients with a higher chance to respond and benefit from therapy. Given the rarity of the disease, there is an immense need for a multi-institutional collaboration alongside industrial support, as well as patient advocacy, to develop the second stage of treatment of penile cancer in the era of immunotherapy.

## Figures and Tables

**Figure 1 jcm-09-03334-f001:**
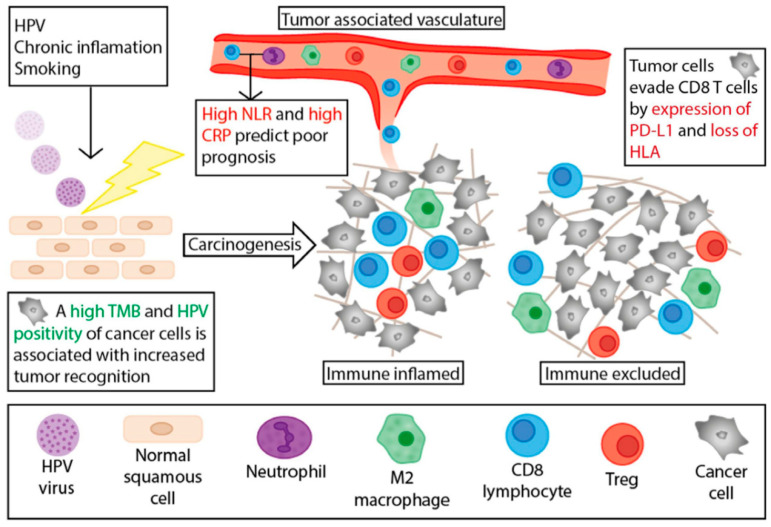
Illustrates tumor microenvironment in penile cancer (Reprinted with permission for Elsevier from “Defining the Tumor Microenvironment of Penile Cancer by Means of the Cancer Immunogram”, by Hielke-Martijn de Vries et al., European Urology Focus, September 2019). Abbreviations: Treg = regulatory T cells, HPV = human papillomavirus; TMB = tumor mutational burden; NLR = neutrophil-to-lymphocyte ratio; CRP = C-reactive protein, PD-L1 = programmed death-ligand 1 [25].

**Table 1 jcm-09-03334-t001:** Recent and ongoing immunotherapy clinical trials in penile cancers.

Study Title	Registration Number	Trial Start Date	Trial Status	Agent	Condition or Disease	Study Outcomes	
						Primary Outcomes	Secondary Outcomes
Phase II Trial of Pembrolizumab for Advanced Penile Squamous Cell Carcinoma Following Previous Chemotherapy	NCT02837042	October 2016	Terminated	Pembrolizumab	Advanced penile squamous cell carcinoma following prior chemotherapy	Objective response rate	- Duration of response - Progression-free survival - Overall survival - Number of adverse events
A Phase 2, Multi-centre, Open-label Study of Avelumab (MSB0010718C) in Locally Advanced or Metastatic Penile Cancer Patients Unfit for Platinum-based Chemotherapy or Progressed on or after Platinum-based Chemotherapy	NCT03391479	15 August 2018	Recruiting	Avelumab	Locally advanced or metastatic penile cancer who are unfit for or progressed on platinum-based chemotherapy	Objective Response Rate	- Progression-free Survival Rate - Overall Survival Rate
A Phase 1 Study of Cabozantinib Plus Nivolumab (CaboNivo) Alone or in Combination with Ipilimumab (CaboNivoIpi) in Patients with Advanced/Metastatic Urothelial Carcinoma and Other Genitourinary Tumors	NCT02496208	9 July 2015	Recruiting	Cabozantinib S-malate plus Nivolumab, plus/minus Ipilimumab	Advanced/Metastatic Genitourinary Tumors	- Recommended dose - Incidence of adverse events	- Clinical response rate - Fraction of alive and progression-free patients at two months - PDL-1 and MET expression
A Phase II Study of Nivolumab Combined with Ipilimumab for Patients with Advanced Rare Genitourinary Tumors	NCT03333616	28 December 2017	Recruiting	Ipilimumab plus Nivolumab	Advanced Rare Genitourinary Tumors	- Objective Response Rate	- Duration of Response - Immune-related objective response rate - Progression-Free Survival - Overall Survival - Safety and tolerability
Phase II Study for the Evaluation of Efficacy of Pembrolizumab (MK-3475) in Patients with Rare Tumors	NCT02721732	15 August 2016	Recruiting	Pembrolizumab	Rare tumors, including metastatic and stage 4 penile cancer	- Non-progression rate - Incidence of adverse events	- Objective response (CR/PR) rates - Duration of response - Progression-free survival - Overall survival - Safety and tolerability
DART: Dual Anti-CTLA-4 and Anti-PD-1 Blockade in Rare Tumors	NCT02834013	13 January 2017	Recruiting	Nivolumab plus Ipilimumab	Rare tumors, including squamous cell carcinoma of the penis	- Overall response rate (ORR)	- Safety and toxicities - Clinical benefit rates - Overall Survival (OS) - Progression-Free Survival (PFS)
PERICLES (Penile Cancer Radio- and Immunotherapy Clinical Exploration Study)-a Phase 2 Study of Atezolizumab With or Without Radiotherapy in Penile Cancer	NCT03686332	25 September 2018	Recruiting	Atezolizumab plus/minus Radiotherapy	Advanced penile cancer	- Progression-free survival at 1 year	- 2-year overall survival rate of the complete study population
The LATENT Trial: Lytic Activation to Enhance Neoantigen-directed Therapy A Study to Evaluate the Feasibility and Efficacy of the Combined Use of Avelumab with Valproic Acid for the Treatment of Virus-associated Cancer	NCT03357757	7 February 2018	Recruiting	Avelumab plus Valproic Acid (VPA)	Human papilloma virus-associated cancers	- 1 year treatment Efficacy - Proportion of patients who complete four doses of Avelumab plus VPA	- Overall survival - Progression-free survival - Adverse events - Identify virus-related cancers for future studies
A Phase II, Single-Arm Open-Label Study of the Combination of Atezolizumab and Bevacizumab in Rare Solid Tumors	NCT03074513	3 March 2017	Active, not recruiting	Atezolizumab, plus Bevacizumab (atezo bev)	Rare solid tumors, including penile squamous cell carcinoma	- Objective response rate (ORR)	- ORR (iRECIST) - Duration of response (RECIST, iRECIST) - PFS (RECIST, iRECIST) - Overall survival
A Phase 2, Open-Label Study to Evaluate Efficacy of Combination Treatment with MEDI0457 (INO-3112) and Durvalumab (MEDI4736) in Patients with Recurrent/Metastatic Human Papilloma Virus-Associated Cancers	NCT03439085	14 November 2018	Recruiting	MEDI0457 (INO-3112), plus Durvalumab (MEDI4736)	Human papilloma virus-associated cancers	- Overall Response Rate (ORR); measured by RECIST	- ORR (iRECIST) - PFS (RECIST) - Disease control rate - Overall survival - Adverse events
Phase II Trial of M7824 in Subjects with HPV Associated Malignancies	NCT03427411	27 February 2018	Recruiting	M7824	Human papilloma virus-associated cancers	- Overall Response Rate (ORR); measured by RECIST	- Duration of response - PFS - OS - Safety and tolerability

Abbreviations: CR = complete response, PR = partial response, ORR = objective response rate, PFS = progression-free survival, RECIST = Response evaluation criteria in solid tumors, iRECIST = immune-based response evaluation criteria in solid tumors, OS = overall survival.

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
