# Peer review of "The Immune Microenvironment in Penile Cancer and Rationale for Immunotherapy"

_jcm, 2020, doi:10.3390/jcm9103334_

Round 1
Reviewer 1 Report
This is a very well-written review by Ahmed and colleagues tackling immune microenvironment in penile cancer. I only have few minor comments:
1- All abbreviations should be revised and defined at their first use. In Figure 1 for example, authors used abbreviations without defining them in the legend.
2- Table 1 is very interesting as it lists the recent and ongoing immunotherapy clinical trials in penile cancers. I suggest authors add the start date of each trial and whether the trial was completed or not yet.
3- I am wondering if there is any literature on desmoplastic stromal reaction in penile tumors. If so, it would be interesting to tackle this point.
4- It is well established that cancer stem cells play an important role in the initiation, progression, and recurrence of many tumors (doi: 10.3389/fnmol.2019.00131) including penile tumors (doi: 10.1038/s41467-020-15980-9). Specifically, Sox2 was shown to be a master regulator for cancer stem cells in skin SCC. Authors are advised to add a paragraph tackling this point.
5- Authors listed some of the common risk factors of penile factors (Introduction section) such as phimosis, smoking, and number of sexual partners. I suggest authors be more specific such as mentioning “higher number of sexual partners” as this is what really refers to the increased risk.
6- Some references that the authors are wrongly cited. For instance, reference [9] is not correct. The paper which talks about therapeutic options being limited with a 5-year overall survival of around 65-70% is not reference [9] but “Verhoeven RH, Janssen-Heijnen ML, Saum KU, Zanetti R, Caldarella A. et al.Population-based survival of penile cancer patients in Europe and the United States of America: no improvement since 1990. Eur J Cancer. 2013;49(6):1414–21. doi: 10.1016/j.ejca.2012.10.029” which is cited within the manuscript of reference [9]. Authors need to revise all the other references thoroughly and make sure to cite original citations correctly.
Author Response
Please see the attachment
Reviewer: 1
This is a very well-written review by Ahmed and colleagues tackling immune microenvironment in penile cancer.
I only have few minor comments:
1- All abbreviations should be revised and defined at their first use. In Figure 1 for example, authors used abbreviations without defining them in the legend.
- Thank you so much for pointing this out. We updated our figure and table with defining the all included abbreviations (Figure1: page 3; lines 91-92; Table1: page 5, lines 126 -128).
2- Table 1 is very interesting as it lists the recent and ongoing immunotherapy clinical trials in penile cancers. I suggest authors add the start date of each trial and whether the trial was completed or not yet.
- Thank you so much for this great suggestion. We agree with the reviewer that including trial start date and recruiting status would be very beneficial for the readers, as such we included these information in table 1 (Page 4-5)
3- I am wondering if there is any literature on desmoplastic stromal reaction in penile tumors. If so, it would be interesting to tackle this point.
While we agree with the reviewer that this is a very interesting area and worth investigation, unfortunately, there is not enough evidence in the literature on the desmoplastic stromal reaction in penile squamous cell carcinoma tumors.
4- It is well established that cancer stem cells play an important role in the initiation, progression, and recurrence of many tumors (doi: 10.3389/fnmol.2019.00131) including penile tumors (doi: 10.1038/s41467-020-15980-9). Specifically, Sox2 was shown to be a master regulator for cancer stem cells in skin SCC. Authors are advised to add a paragraph tackling this point.
- Thank you for this excellent suggestion. We have included a paragraph describing the recent finding on the role of cancer stem stems and their transcriptional regulators in SCC initiation and disease progression (Page 8, lines 304-311)
“Additionally, recent reports have described the upregulation of Sox2 and Ptgs2 transcriptional factors in skin and penile SCC [57,97]. Sox2 is a β-catenin transcriptional target, upregulated in cancer stem cells, and absent in normal epidermis, while Ptgs 2, or better known as Cox2, is a pro-inflammatory gene that induces prostaglandins production from arachidonic acid resulting in cytokine upregulation and inflammatory reactions. These genes act as the master regulators of cancer stem cells that play an important role of Sox-2 in tumor initiation and progression. Boumahdi et al. reported that conditional deletion of Sox2-gene results in tumor cell regression, consistent with their critical role for disease continuity [97].”
5- Authors listed some of the common risk factors of penile factors (Introduction section) such as phimosis, smoking, and number of sexual partners. I suggest authors be more specific such as mentioning “higher number of sexual partners” as this is what really refers to the increased risk.
- Thank you for pointing this out. We agree with the reviewer and edited our introduction specifying that “high” number of sexual partner is the risk factor for penile cancer (page 1, line 33)
6- Some references that the authors are wrongly cited. For instance, reference [9] is not correct. The paper which talks about therapeutic options being limited with a 5-year overall survival of around 65-70% is not reference [9] but “Verhoeven RH, Janssen-Heijnen ML, Saum KU, Zanetti R, Caldarella A. et al.Population-based survival of penile cancer patients in Europe and the United States of America: no improvement since 1990. Eur J Cancer. 2013;49(6):1414–21. doi: 10.1016/j.ejca.2012.10.029” which is cited within the manuscript of reference [9]. Authors need to revise all the other references thoroughly and make sure to cite original citations correctly.
- Thanks for the reviewer for pointing out our mistake, and we apologize for citing the wrong reference. We have checked our reference, and corrected them whenever needed.
Reviewer 2 Report
Ahmed and colleagues summarize the literature regarding the immune microenvironment in penile cancer and provide an overview of selected ongoing clinical trials. The manuscript is well written (some spelling and grammar mistakes aside).
I have some minor comments:
The authors mention some parallels between penile SCC and head and neck SCC. It would be useful to add a brief section in the introduction explaining that in fact penile cancers share different features with other squamous cell cancers regarding stromal and immune alterations. There is an excellent review on this topic which should be cited : Dotto GP & Rustgi A. Cancer Cell. 2016 May 9; 29(5): 622–637.
For Table 1: it is very useful for clinicians interested in the topic, it would be however good to indicate the completion date of the studies in order to have an overview of the ongoing trial. The authors should also mention when they have accessed the clinicaltrials.gov website.
The reference 95 should be corrected, the authors cite the abstract of a study, which is available as full text under Clin Cancer Res. 2015 Mar 1; 21(5): 1196–1206.
Author Response
Reviewer: 2
Ahmed and colleagues summarize the literature regarding the immune microenvironment in penile cancer and provide an overview of selected ongoing clinical trials. The manuscript is well written (some spelling and grammar mistakes aside).
I have some minor comments:
The authors mention some parallels between penile SCC and head and neck SCC. It would be useful to add a brief section in the introduction explaining that in fact penile cancers share different features with other squamous cell cancers regarding stromal and immune alterations. There is an excellent review on this topic which should be cited: Dotto GP & Rustgi A. Cancer Cell. 2016 May 9; 29(5): 622–637.
- We would like to thank the reviewer for his excellent suggestion. We have included a paragraph in our introduction describing similarities in the epigenetic factors and the pathogenesis regulators of SCC from various sites (page 1, lines 34-38)
“While penile SCC has a separate entity of SCCs and treated differently, there are recent reports describing extensive similarities and commonalities in the genetic and pathogenesis regulators of SCCs of various sites, including both the general determinants in the cancer process such as P53 and cyclin D1, or the specific regulators such as NOTCH, SOX2, and TP63 genes [10].”
For Table 1: it is very useful for clinicians interested in the topic, it would be however good to indicate the completion date of the studies in order to have an overview of the ongoing trial. The authors should also mention when they have accessed the clinicaltrials.gov website.
- As addressed in response to a prior point we have now included trial start date and the recruiting status as per 2020 updates. (Page 4-5, lines 125- 129)
The reference 95 should be corrected, the authors cite the abstract of a study, which is available as full text under Clin Cancer Res. 2015 Mar 1; 21(5): 1196–1206.
- We apologize for the reviewer for this mistake and now we have checked our reference, and corrected them whenever needed.